# Effect of Cu Content on the PBF-LB/M Processing of the Promising Al-Si-Cu-Mg Composition

**Alessandra Martucci** [1,*], **Emilio Bassini** [1] and **Mariangela Lombardi** [1,2,*]

1   Department of Applied Science and Technology, Politecnico di Torino, Corso Duca Degli Abruzzi 24, 10129 Turin, Italy; emilio.bassini@polito.it

2   Consorzio Interuniversitario Nazionale per la Scienza e Tecnologia dei Materiali (INSTM), Via G. Giusti 9, 50121 Florence, Italy

*   Correspondence: alessandra.martucci@polito.it (A.M.); mariangela.lombardi@polito.it (M.L.)

**Abstract:** Over the past few years, several studies have been conducted on the development of Al-Si-Cu-Mg alloys for PBF-LB/M processing. The attention gained by these systems can be attributed to their light weight and strength provided by a solid solution in the as-built state and by precipitation after heat treatment. However, published studies have kept the copper content below its solubility limit in the Al-Cu binary system under equilibrium conditions (5.65 wt%). The present study aims to explore Al-Si-Cu-Mg systems with high copper content, starting with the well-known AlSi10Cu4Mg system, moving towards AlSi10Cu8Mg, and arriving at AlCu20Si10Mg, a system never before processed with PBF-LB/M. Through the SST approach, the production of bulk samples, advanced microstructural characterization by SEM and FESEM analysis, phase identification by XRD analysis, and preliminary investigation of the mechanical properties through Vickers micro indentations, the effects of copper quantities on the processability, microstructural properties, and mechanical behavior of these compositions were investigated. The obtained results demonstrated the benefits of the supersaturated solid solution and the fine precipitation resulting from the addition of high Cu contents. In particular, the AlCu20Si10Mg system showed a very distinctive microstructure and unprecedented microhardness values.

**Keywords:** Powder Bed Fusion Processing; PBF-LB/M; Al-based alloys; Al-Si-Cu-Mg; Cu solubility limit; $Al_2Cu$ phase; microstructural characterization

## 1. Introduction

Among the most widespread and investigated alloys of the last century, it is possible to note aluminum alloys, which are characterized by a low specific weight, excellent strength-to-weight ratio, intrinsic corrosion resistance, good thermal and electrical conductivity, and optimal formability and machinability. In particular, alloys based on the Al-Si binary system have been increasingly used for traditional processing, such as casting, especially with quasi-eutectic compositions [1]. These compositions allow reduced shrinkage, high weldability, and a low melting range imparted by a large volume fraction of eutectic Al-Si. However, the mechanical properties of these alloys hardly meet the automotive industry goals of obtaining lightweight materials with high specific strength, as they are characterized by moderate strength and hardness and low toughness in the as-processed state [1]. In addition, Al-Si alloys are poorly suited to heat treatment as, with increasing temperature, the Si in solid solution is rejected from the Al matrix to form particles, which rapidly grow in size and number, drastically reducing mechanical properties [2]. In order to enhance mechanical properties, alloying elements such as Mg and Cu are often added [3]. It was, in fact, proved that a small amount of Mg (0.3–0.5 wt%) added to the Al-Si alloy could significantly improve the strength after heat treatment thanks to the precipitation of dispersed $Mg_2Si$ nanoparticles [4]. Furthermore, the addition of Cu can further increase

the age-hardening kinetics of Al-Si-Mg alloys through the precipitation of the quaternary Q phase and θ (Al$_2$Cu) phase [5].

Despite the addition of Mg and Cu as alloying elements, the mechanical properties of Al-Si-based alloys processed by conventional processes are strongly affected by the microstructural features, such as the arm spacing of primary α-Al dendrites, the morphology and size of the secondary phases, and the presence of casting defects [6,7]. In particular, when high quantities of copper are present in the alloy, the slow cooling rates typical of conventional processes lead this element to segregate at grain boundaries. Cu segregation and precipitation of large θ particles must be controlled because these phenomena could interfere with the solid-state diffusion at the base of precipitation hardening heat treatments. In addition, even after long solution heat treatments, such segregations cannot be wholly dissolved, leading to alloy embrittlement [8]. To avoid these detrimental phenomena, it is possible to act by adding other modifiers to the alloy [9–11], performing ultrasonic melting treatments [10,12], or significantly increasing the cooling rate during the solidification. Regarding this latter aspect, Al-Si-Cu-Mg alloys can be processed with additive manufacturing processes, which have several-orders-of-magnitude-higher cooling rates compared to conventional processes such as die casting. In particular, during the process of Laser-Based Powder Bed Fusion for Metals (PBF-LB/M), the solidification is achieved with cooling rates up to $10^7$ K/s. This intense cooling rate guarantees an extremely fine microstructure and extended solubility limits [13]. Furthermore, such high cooling rates lead to kinetically modified structures that cannot form under conditions closer to equilibrium. For example, Qin et al. [14] showed that, with the cooling rates achievable in the PBF-LB/M process, the volume fraction of the eutectic phase and the solubility limit of Si in α-Al largely deviate from the equilibrium values determined by the Al-Si equilibrium phase diagram. In addition, as widely observed in the literature and verified by Tang et al. [15], the solidification rate is inversely proportional to the cell size or spacing of the secondary dendrite arms. This justifies the unique fine microstructure achievable in the components manufactured by PBF-LB/M. Rapid solidification also partially solves the issues of coarse solute segregations that characterize slow conventional processes, generating supersaturated microstructures with fine, well-dispersed nanoprecipitates [16]. In addition, the nanometric solute clusters formed during the PBF-LB/M process play a key role in improving the strength and strain hardening capacity of the alloy in the as-built state [17,18]. Microstructural peculiarities resulting from the PBF-LB/M process lead to Al-Si-Cu-Mg alloys with unprecedented mechanical properties [19].

Several attempts have been made over the past few years to process Al-Si-Cu-Mg alloys via PBF-LB/M [20,21]. Most studied systems, according to the literature, generally contain Si values close to eutectic (around 10 wt%), Mg values limited between 0.3 and 0.5 wt%, and copper values within 4 wt% [21–23]. The limited amount of copper in the systems already explored in the literature could be explained by the intention to not exceed the solubility limit of Cu in aluminum, which, in the Al-Cu binary system, is 5.65 wt% [24]. Overcoming this limit can lead to an undesirable extension of the alloy solidification range and thus to the occurrence of hot tearing, as demonstrated by Lu et al. [25]. In addition, as proved by Aksöz et al., copper as an alloying element in Al-based compositions results in decreased thermal conductivity [26]. This reduction worsens heat diffusion during the PBF-LB/M process, consequently changing the process condition to an adequate final densification [27]. A previous work from these researchers addressed the processability via PBF-LB/M of an Al-Si-Cu-Mg alloy exceeding the copper limit of 5.65 wt% [28]. In particular, it was proved that a crack-prone AlSi10Cu8Mg alloy can also be successfully processed via PBF-LB/M with tight control of the processing parameters and the use of support structures. Furthermore, the study conducted on the AlSi10Cu8Mg revealed a mean hardness value of ca. 190 HV, a 22% higher value compared to those published in previous works by Martin et al. and Martucci et al. with the AlSi10Cu4Mg system [21,22]. The positive preliminary results obtained with the AlSi10Cu8Mg alloy pave the way for Al-Si-Cu-Mg alloys with higher copper content studies.

The present study provided an important opportunity to advance the understanding of the effects of the amount of copper in Al-Si-Cu-Mg alloys on the PBF-LB/M processability, microstructural evolutions, and microhardness. The well-known AlSi10Cu4Mg system was used in the present work as a benchmark, and the AlSi10Cu8Mg and AlCu20Si10Mg systems were carefully investigated in order to fill the literature gap regarding high-copper systems processed by PBF-LB/M. The effects of the copper quantities on processability were investigated using a PBF-LB/M lab-scale machine, firstly with a preliminary fine-tuning through the study of single scan tracks (SSTs) and then with the production of massive samples. A preliminary investigation of the mechanical properties was then carried out through Vickers micro indentations to explore the potential of these compositions. The microstructural characteristics were investigated by advanced observation with Scanning Electron Microscopy (SEM) and Field Emission Scanning Electron microscopy (FESEM) analyses, using different detectors on etched samples with various reagents. Finally, phase identification was performed with X-Ray Diffraction (XRD) analysis.

## 2. Materials and Methods

### 2.1. Pre-Alloyed Powder Characterization

To understand the copper role in the PBF-LB/M processability, microstructural features, and mechanical behavior of Al-Si-Cu-Mg systems, in the present study, AlSi10Cu4Mg, AlSi10Cu8Mg, and AlCu20Si10Mg compositions were selected. Our research group utilized a PSI HERMIGA100/10 VI gas atomizer (Phoenix Scientific Industries Ltd., Hailsham, East Sussex, UK) to produce the abovementioned powder systems, starting from the AlSi10Mg system and progressively adding pure elements to obtain the desired stoichiometric ratios. Before the vacuum induction melting, a backfill process was performed to prevent elements with a low boiling point (or vapor pressure), i.e., Mg, from evaporating. To avoid problems related to oxidation, the entire gas-atomization process was conducted in an atmosphere of high-purity argon gas. Finally, a passivation step was implemented after gas atomization to prevent explosive situations caused by the high powder reactivity.

The system classification and the relative chemical compositions are reported in Table 1.

**Table 1.** Chemical composition of gas-atomized systems.

| Composition Range (wt%) | Al | Si | Cu | Mg |
|---|---|---|---|---|
| AlSi10Cu4Mg | Balance | ~10 | ~4 | ~0.5 |
| AlSi10Cu8Mg | Balance | ~10 | ~8 | ~0.5 |
| AlCu20Si10Mg | Balance | ~10 | ~20 | ~0.5 |

Before checking the particle size and morphology, the powders were sieved in the 20–50 μm range, as required for PBF-LB/M production. The particle size distribution (PSD) was measured by the Mastersizer3000 (Malvern Panalytical, Malvern, UK) laser diffraction particle size analyzer, and the particle morphology was checked by SEM images taken by Phenom XL SEM (Thermo Fisher Scientific, Waltham, MA, USA). To observe the distribution of the alloying elements, the powder microstructures were then explored through SEM analyses, exploiting the back-scattered electron (BSE) detector and energy dispersive X-ray (EDS) analyses with a Phenom ProX (Thermo Fisher Scientific, Waltham, MA, USA). Finally, the apparent density ($\rho a$) and tap density ($\rho t$) of each powder system were measured three times according to ASTM B417-18 and ASTM B527-15, respectively. The Hausner ratio (H) was then calculated as the ratio of the tap and apparent densities as H = $\rho t / \rho a$. Before the PBF-LB/M production, the sieved powder systems were dried at 90 °C for two hours. Based on Weingarten et al.'s evidence, the powder-drying procedure allows a level of gas pores after PBF-LB/M production to be markedly reduced [29]. Powder reflectance measurements were performed with the ultraviolet/visible light (UV/Vis) spectrophotometer Varian Cary 5000 (Agilent Technologies, Leini, Turin, Italy) at a wavelength ranging from 400 to 1500 nm in order to investigate the effect of different amounts of copper on the PBF-LB/M

processability of the alloys analyzed. The wavelength range was chosen to include the specific wavelength of the YAG laser used for the PBF-LB/M production (1070 nm). The powder samples were placed in a container and closed with a glass plate for the integrating sphere attachment.

*2.2. PBF-LB/M Production*

For the entire PBF-LB/M production, a laboratory-scale Concept Laser Mlab Cusing R (General Electric Company, Boston, MA, USA) equipped with a gaussian beam fiber laser with a maximum power of 100 W, a wavelength of 1070 nm, and a laser spot of 50 μm was used. The PBF-LB/M system featured a build volume of $90 \times 90 \times 80$ mm$^3$. Specimens were built onto a pure Al platform without preheating. During the PBF-LB/M process, high-purity Ar gas ($\geq$99.99%) was fed into the building chamber, decreasing the $O_2$ content to below 0.2%. To identify the power (*P*) and scan speed (*v*) values suitable for the bulk production and minimizing the time and powder needed, the SST approach was used. In order to further accelerate this first fine-tuning stage of the *P* and *v* parameters, a software developed by the authors in a previous work was used [30]. This software is able to identify the most suitable $P - v$ combinations by discarding the noncontinuous scans and considering the SST quality just with a fast image analysis of the on-top morphology. This tool was exploited by analyzing a wide range of power and scan speed values (*P* = 80–95 W, and *v* = 300–1400 mms$^{-1}$), resulting in a Linear Energy Density (*LED*) range from 0.09 to 0.14 Jmm$^{-1}$, applying Equation (1):

$$LED = \frac{P}{v} \tag{1}$$

After finding the most suitable *P-v* combinations for the three systems, a third crucial parameter for bulk production, the hatch distance (*hd*), was evaluated. According to the methodology implemented by Bosio et al. [31], *hd* values were set by considering the mean SST width and an overlapping of 0% between nearby tracks. In addition, following the standard indications of PBF-LB/M machine suppliers for Al-based alloys, the layer thickness (l) was fixed at 15 μm in each system. Finally, a chessboard scanning strategy with a rotation of 90° was used as the scanning strategy for the entire bulk production in order to generate an isotropic stress tensor, reducing residual stresses in the samples [32]. To understand the effect of the PBF-LB/M process parameter on the final densification level, the volumetric energy density (*VED*) index was used. This index embodies the main PBF-LB/M process parameters according to the Equation (2):

$$VED = \frac{P}{v \, hd \, l} \tag{2}$$

The 10 mm$^3$ cubic samples realized through the use of the first process parameters identified from the SST analysis revealed a strong delamination and crack formation for the systems with higher Cu content, AlSi10Cu8Mg and AlCu20Si10Mg. To avoid delamination and crack formation, AlSi10Cu8Mg and AlCu20Si10Mg were processed, exploiting lower scan speed values and support structures. This approach was validated by the authors in a previous work on process optimization for crack-prone alloys [28]. A fully circular (conical and cylindrical) configuration was selected as the basic geometry of the support structures. This geometric configuration has already been explored by Han et al. [33], revealing an excellent ability to anchor overhang structures and an easy-to-model heat transmission. The support heights and diameters were set according to the authors' previous optimization work performed on AlSi10Mg samples. The geometry of support structures and their arrangement under the cubic specimens are illustrated in Figure 1.

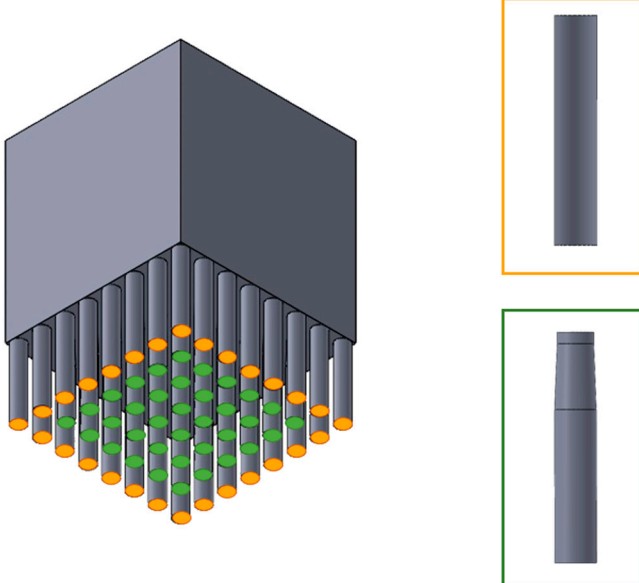

**Figure 1.** Geometries of support structures and their location under the samples.

*2.3. As-Built Samples Characterization*

After the PBF-LB/M process, the as-built specimens were cut from the building platform using an electrical discharge machine. To assess the densification level and identify potential internal cracks, all samples were cut parallel to the build direction and polished following the standard metallographic procedure until their cross-sections were mirror-finished. When necessary, the samples were embedded in conductive resin before the cutting and polishing procedure. Image-processing software, ImageJ, was used to calculate the densification level. In detail, a Leica DMI 5000 M (Leica Camera AG, Wetzlar, Germany) optical microscope was used to take micrographs ($100\times$ magnification) covering the entire sample cross-section. Subsequently, all the images for each sample were processed using ImageJ software, which applied thresholding to distinguish pores and cracks from the matrix and calculate the relative density. By analyzing the densities obtained with the different parameter combinations suggested by the SST approach, the best process conditions were identified for each system. The samples of each system that revealed the highest relative density were accurately investigated to determine their processability and microstructural and mechanical properties. For an initial understanding of the PBF-LB/M processability of the three systems, the general internal defect condition was investigated using a Leica EZ4 W stereomicroscope (Leica Camera AG, Wetzlar, Germany). To discover the potential of the three systems, a preliminary investigation of their mechanical properties was performed through micro indentations, using a Vickers tester, VMHT Leica (Leica Camera AG, Wetzlar, Germany). The measurements were carried out according to ASTM E384 standard by performing ten microhardness indentations on the XZ plane, using a static load of 0.5 kg and a dwell time of 10 s. The influence of the Cu amount in the alloy on the hardness values was then supported by the extensive investigation of the microstructural features of the three systems. In order to obtain an overview of the microstructural characteristics and perform a compositional map to observe the distribution of the alloying elements, a thorough investigation with the ZEISS EVO 15 SEM (Zeiss, Oberkochen, Germany) equipped with an EDS detector was carried out on the samples chemically etched with the Keller reagent. In order to obtain a great contrast between the Al matrix and the $Al/Si/Al_2Cu$ network, the samples were then chemically etched with the Tucker reagent, which mainly etched the Al matrix and slightly etched the Al-Si eutectic phase, highlighting the $Al_2Cu$ phase in the network well [34]. To obtain an accurate microstructural analysis at high magnifications, the samples were then observed with a Field Emission Scanning Electron Microscope (FESEM), FESEM Merlin Zeiss SEM (Zeiss, Oberkochen, Germany). Phase identification was conducted using

a PANalytical X-Pert diffractometer (Malvern Panalytical, Malvern, UK). XRD analyses were performed with a Bragg Brentano configuration, employing a CuK$\alpha$ radiation at 40 kV and 40 mA. Reference patterns, including Star 00-004-0787 for aluminum (Al), Calculated 01-089-1980 for Al$_2$Cu, and Calculated 01-077-2109 for silicon (Si), were utilized for the phase identification procedure. To determine the lattice parameter of face-centered cubic (fcc) Al, the cos$\theta$ cot$\theta$ method was employed. The quantification of the supersaturation condition in the Al matrix involved the application of Vegard's law, utilizing the lattice parameter of the PBF-LB/M sample and the near-equilibrium lattice parameter. The near-equilibrium behavior was established by examining a sample subjected to melting, followed by a markedly slow cooling process to achieve a near-equilibrium microstructure. The reaction sequence was determined through a DSC analysis performed with a NETZSCH DSC 214 Polyma instrument (NETZSCH Group, Waldkraiburg, Germany), using an Al$_2$O$_3$ crucible and a protective argon atmosphere. The heating rate was set at 10 °C/min, and the temperature range spanned from 20 to 450 °C.

## 3. Results and Discussion

### 3.1. Powders Assessment: Morphology and Chemistry

Metal powders must satisfy specific requirements for a successful processing by PBF-LB/M. In particular, the particles need to be characterized by a spherical shape since, as stated by Baitimerov et al. [35], non-spherical particles lead to a low apparent density of the powder system and, thus, a potential non-optimal densification after the PBF-LB/M process. In addition, a limited presence of satellites is required to ensure a proper powder flowability and, thus, to achieve a higher quality of the powder bed. A key role in the packing and flow properties of powders is also played by the PSD, which must find the right balance between fine and coarse particle fractions [36]. To verify the characteristics of the three gas-atomized systems, powders in the 20–63 µm fraction were observed by SEM. As can be observed in Figure 2, all systems are characterized by spherical particles with a limited number of satellites and narrow PSDs with D50 values of 41.2, 38.3, and 37.3 µm for AlSi10Cu4Mg (Figure 2a), AlSi10Cu8Mg (Figure 2b), and AlCu20Si10Mg (Figure 2c), respectively, meeting the requirements of the PBF-LB/M production.

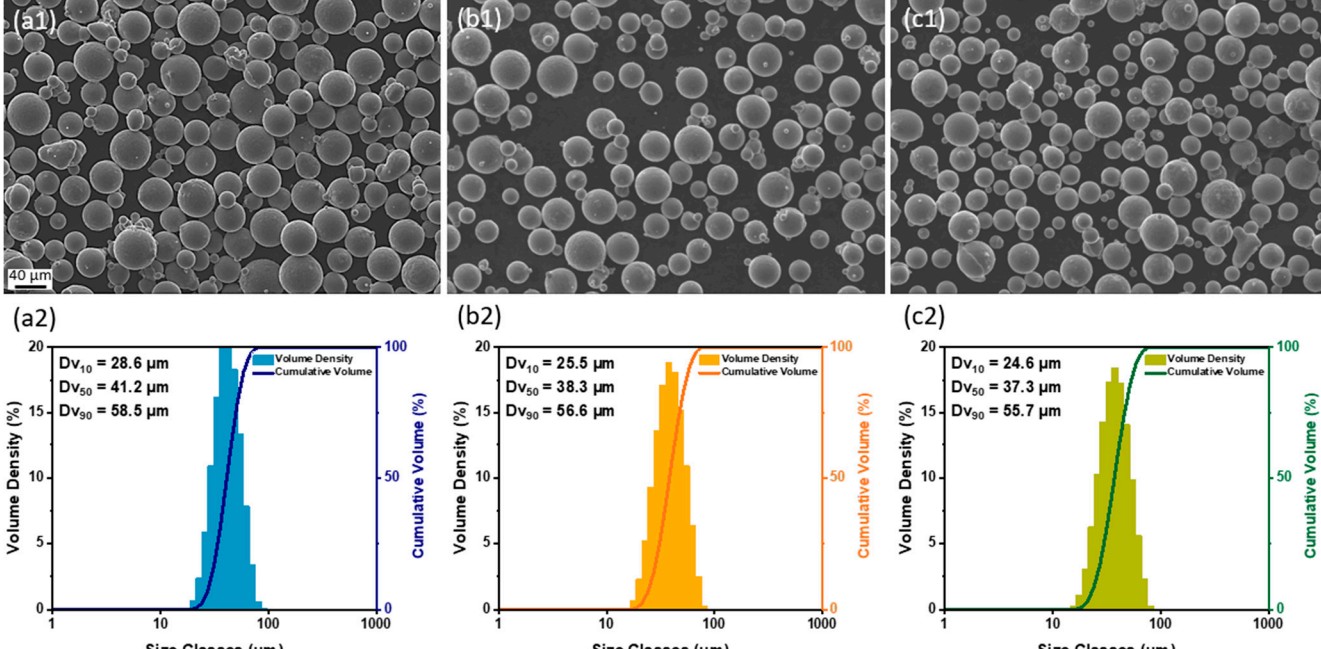

**Figure 2.** SEM powder micrographs with relative PSDs of AlSi10Cu4Mg (**a1**,**a2**), AlSi10Cu8Mg (**b1**,**b2**), and AlCu20Si10Mg (**c1**,**c2**).

In order to verify if the three systems are subjected to flow and present packing properties suitable for PBF-LB/M processing, their H ratios were calculated based on their tap density and apparent density values. In fact, a higher H indicates poorer flowability and packing characteristics of the powder bed, suggesting that the particles are loosely packed and prone to interparticle friction, resulting in difficulties in powder flow. Conversely, a lower H ratio indicates better flowability and improved packing properties, indicating that the particles are tightly packed, with reduced interparticle friction [37]. In particular, Sutton et al. [38] proved that when an H ratio of less than 1.25 is achieved, the powder can be considered suitable for the PBF-LB/M process. The AlSi10Cu4Mg, AlSi10Cu8Mg, and AlCu20Si10Mg systems revealed a Hausner ratio of 1.11, 1.09, and 1.10, respectively. According to Sutton et al. [38], it is therefore possible to state that the three systems should exhibit a good level of packability and flowability during the additive process. Furthermore, no significant variations in the flow and packing behavior of the systems can be discerned from the measured H values.

To investigate the effects of the copper quantity on the particle microstructure, the powders were embedded in conductive resin and polished until their cross-sections were suitable for SEM investigation with the BSE detector and EDS probe (Figure 3).

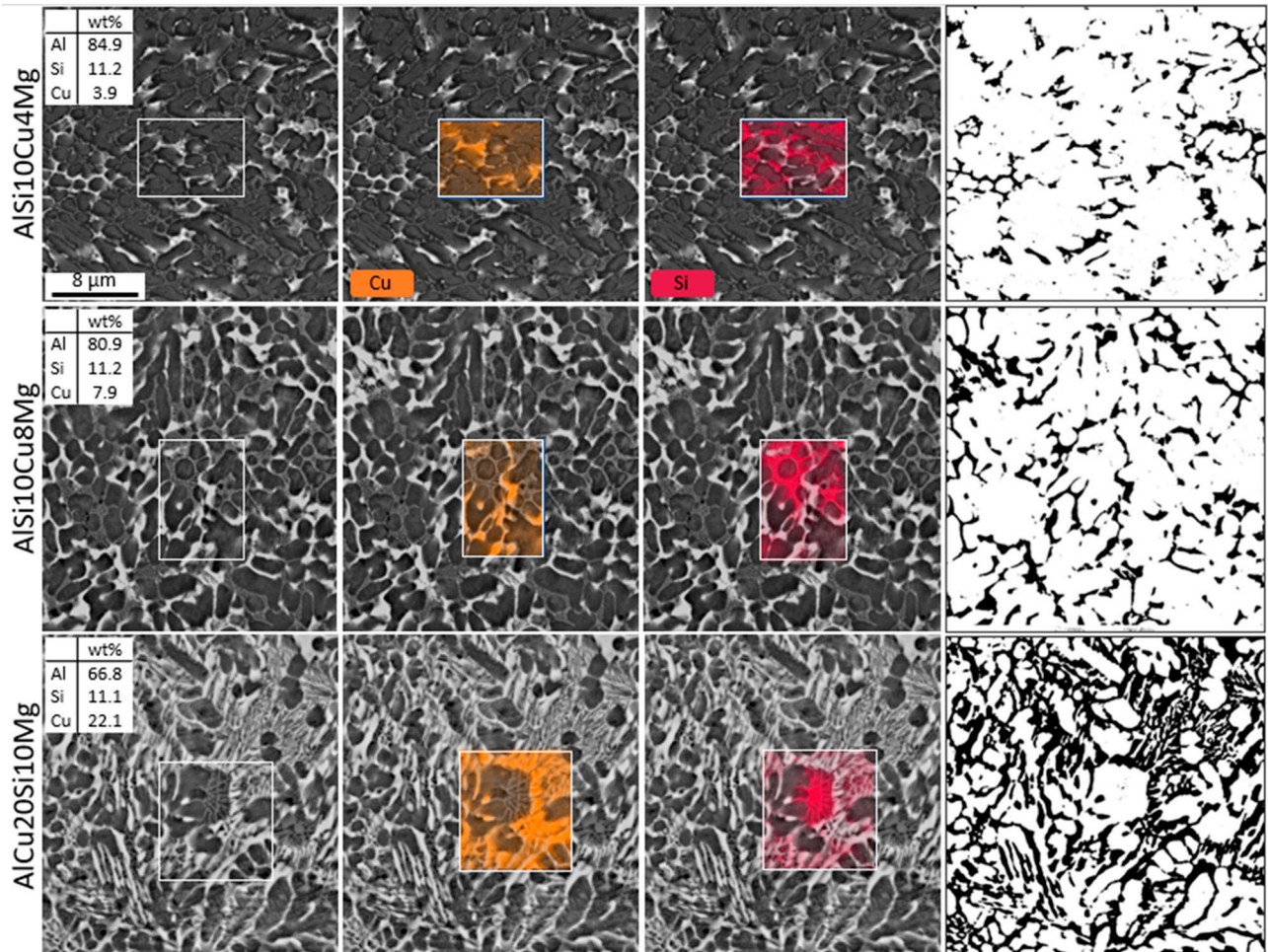

**Figure 3.** Cross-sectional micrographs of AlSi10Cu4Mg, AlSi10Cu8Mg, and AlCu20Si10Mg powders observed by SEM with BSE detector, EDS analyses, and relative threshold maps obtained with ImageJ.

Looking at cross-section micrographs in Figure 3, we can see that the particle microstructures appear to be characterized by a matrix and a cellular network consisting of two different phases: one with a higher atomic number (brighter in the image) and one with a lower atomic number (darker in the image). In order to identify the predominant

elements in the different network zones, EDS maps were conducted on each system. The EDS results, reported in Figure 3, clearly demonstrate that the brighter areas revealed by the SEM investigation were enriched in Cu, and the darker ones in Si. Based on the literature related to similar compositions, it is reasonable to assume that the matrix is the α-Al phase and the network is composed of the Al-Si eutectic and the $Al_2Cu$ phase [21,22,39]. The microstructures of the AlSi10Cu4Mg and AlSi10Cu8Mg systems appear morphologically comparable, but by applying contrast image analysis with ImageJ software, it was possible to assess a fraction of $Al_2Cu$-rich areas in the AlSi10Cu8Mg that was 7% higher than in the AlSi10Cu4Mg. On the contrary, the micrograph of AlCu20Si10Mg reveals a distinctive morphology. In particular, the network appears markedly coarsened, and some areas of the network reveal a Cu- and Si-rich lamellar structure not found in other systems. Although the gas-atomization process deviates from equilibrium conditions, the ternary Al-Si-Cu diagram can be used to obtain an indication of the reason why these important microstructural differences occur [40].

Based on the equilibrium ternary Al-Si-Cu phase diagram in Figure 4, it is, in fact, possible to state that, in the AlCu20Si10Mg system, the first phase to solidify could be Si, in contrast to the other two lower Cu systems, where the first phase to be created could be Al. However, as stated before, the divergence from equilibrium conditions due to the rapid solidification involved in the gas-atomization process does not make the solidification sequence precisely identifiable [41]. Moreover, the presence of Mg in the alloy further complicates the phase diagram. Notwithstanding, the morphological similarity of the AlSi10Cu4Mg and AlSi10Cu8Mg systems suggests that similar solidification sequences characterize them. In contrast, the considerable morphological differences of the AlCu20Si10Mg system indicate that it is characterized by completely different solidification sequences.

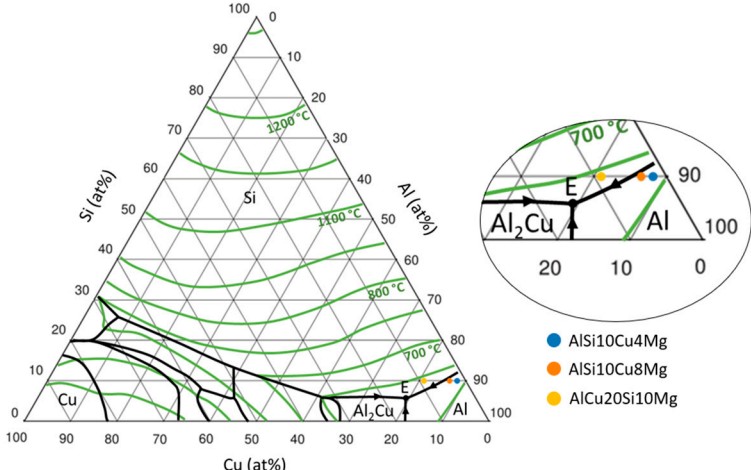

**Figure 4.** Al-Si-Cu phase diagram in equilibrium condition (based on the study conducted by Hallstedt et al., with permission from Ref. [40]. Copyright 2023 Elsevier).

### 3.2. Effect of Copper on PBF-LB/M Processability

The laser–powder interaction in the PBF-LB/M process is of paramount importance, as it governs the melting, fusion, energy transfer, melt pool dimensions, and overall solidification behavior [42]. Specifically, the laser beam interacts with the powder particles through a process called absorption, in which the powder material absorbs the laser energy and converts it into heat. Aluminum and copper demonstrated a poor absorption rate and, thus, were highly reflective to near-infrared laser wavelengths commonly used in PBF-LB/M, i.e., 1070 nm as the laser used in the present study [43,44]. The reflective nature of these materials causes a significant portion of the laser energy to bounce off the surface rather than being absorbed by the powder particles. Consequently, a significantly high energy density could be required to effectively melt an Al-based composition with Cu. In order to investigate the effect of increasing amounts of alloyed copper on laser absorbance,

UV-Vis analyses were conducted. The results obtained for the three systems are displayed in Figure 5.

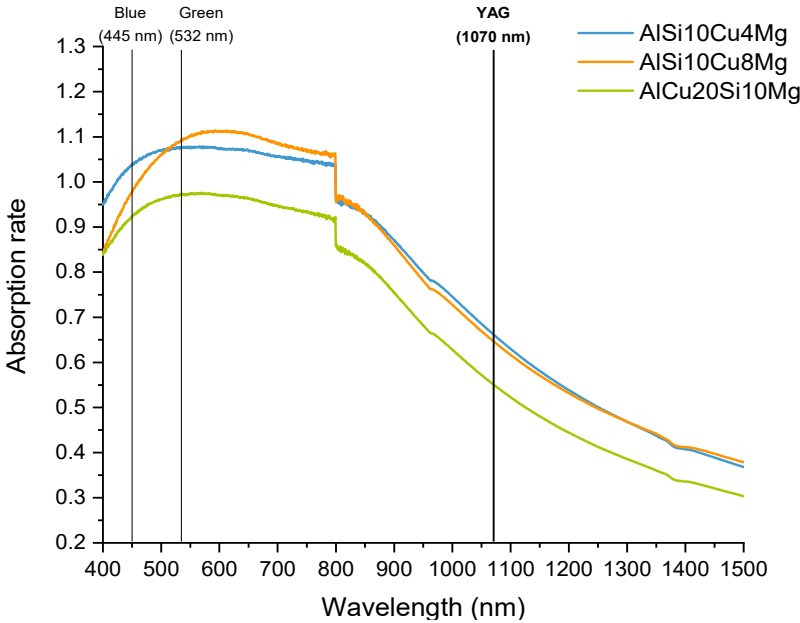

**Figure 5.** The absorption of laser output at different wavelengths for the AlSi10Cu4Mg, AlSi10Cu8Mg, and AlCu20Si10Mg systems. The drop is due to the lamp change during the test.

The trends in Figure 5 reveal that, at the wavelength of the YAG laser used for PBF-LB/M production, the systems AlSi10Cu4Mg and AlSi10Cu8Mg are characterized by comparable absorbance levels. On the contrary, the higher amount of copper in the AlCu20Si10Mg system significantly impairs its laser absorption. The results obtained suggest that the AlCu20SiMg system could require more energy-intensive process parameters to be properly processed and densified compared to the AlSi10Cu4Mg and AlSi10Cu8Mg systems. However, although the UV-Vis analysis can provide an indication of the laser–powder interaction during the PBF-LB/M, the quality of the powder bed significantly affects the powder behavior during the laser scans [45]. In fact, more void spaces and gaps between the particles occur in a loosely packed or low-density powder bed. When the laser beam travels through the powder bed and encounters these voids, the laser scattering increases, causing a dispersion of the laser energy and a decrease in the beam intensity. To obtain a more accurate idea of the laser–powder interaction during the PBF-LB/M processing of the three systems, it is necessary to investigate the quality of the powder bed; therefore, an analysis of the flow and packing properties of the powder was performed. The H ratio results can be helpful in identifying possible critical points in the laying of the powder bed. Considering that the three systems recorded very similar H values that are compatible with a good powder bed quality, it can be assumed that the UV-Vis test well reflects the behavioral differences of the three systems during the PBF-LB/M process.

To quickly identify promising process parameters for an adequate PBF-LB/M bulk densification, the SST approach, based on evaluating their on-top morphology, was explored. For all systems, the same wide ranges of power and scan speed values were explored, and the results obtained after the SST software evaluation are reported in Figure 6. In particular, in Figure 6, the conditions revealed strong discontinuities are marked in red, and the conditions that the software identified as discardable due to a highly irregular SST morphology are indicated in orange.

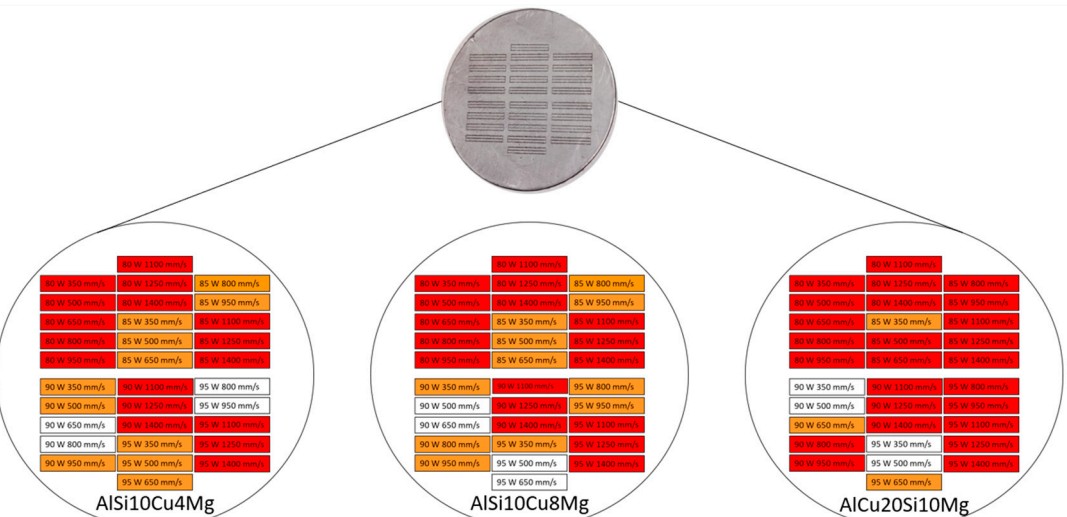

**Figure 6.** The SST parameters explored for a preliminary fine-tuning of the PBF-LB/M process parameters.

Looking at the conditions not discarded by the software (left in white in Figure 6), it is immediately evident that the higher the amount of copper in the alloy, the lower the scan speed values required and, consequently, the higher the LED. In detail, the parameters suggested for the AlSi10Cu8Mg system were slightly more energy-intensive than those for the AlSi10Cu4Mg system. In contrast, the AlCu20Si10Mg system revealed a need for significantly lower scan speeds and, thus, a considerably higher energy density than the lower-copper systems. These results are perfectly in line with what the absorbance analysis revealed (Figure 5). The higher amount of copper leads to lower absorption rates and, thus, to more energy-intensive process parameters for the proper processing and densification.

Exploiting *P-v* combinations not discarded by the SST software, following Concept Laser indications for the layer thickness values suitable for Al-based alloys, and setting hd values calculated based on the SSTs width, four cubic bulk samples for each system were processed via PBF-LB/M. In order to identify the process condition for the bulk production, the relative densities were evaluated through image analysis. The results obtained are reported in Table 2.

**Table 2.** PBF-LB/M process parameters based on SST results used for the bulk production. In green are the parameters that revealed the highest relative density.

|  | Power (W) | Scan Speed (mms$^{-1}$) | Hatch Distance (μm) | Layer Thickness (μm) | VED (Jmm$^{-3}$) | Relative Density (%) |
|---|---|---|---|---|---|---|
| AlSi10Cu4Mg | 90 | 650 | 97 | 15 | 95 | 98.74 |
|  | 90 | 800 | 92 | 15 | 81 | 96.78 |
|  | 95 | 800 | 85 | 15 | 93 | 97.55 |
|  | 95 | 950 | 80 | 15 | 83 | 97.31 |
| AlSi10Cu8Mg | 90 | 500 | 112 | 15 | 107 | 82.45 |
|  | 90 | 650 | 106 | 15 | 87 | 80.45 |
|  | 95 | 500 | 117 | 15 | 108 | 83.75 |
|  | 95 | 650 | 115 | 15 | 85 | 80.12 |
| AlCu20Si10Mg | 90 | 350 | 120 | 15 | 143 | 93.25 |
|  | 90 | 500 | 103 | 15 | 116 | 91.25 |
|  | 95 | 350 | 125 | 15 | 145 | 93.78 |
|  | 95 | 500 | 118 | 15 | 107 | 90.87 |

Among the four parameter combinations investigated for each composition, it was decided to pursue the one that revealed a higher relative density from the image analyses (marked in green in Table 2). The micrographs of the three systems after the PBF-LB/M process, with the parameters ensuring the highest densification, are displayed in Figure 7.

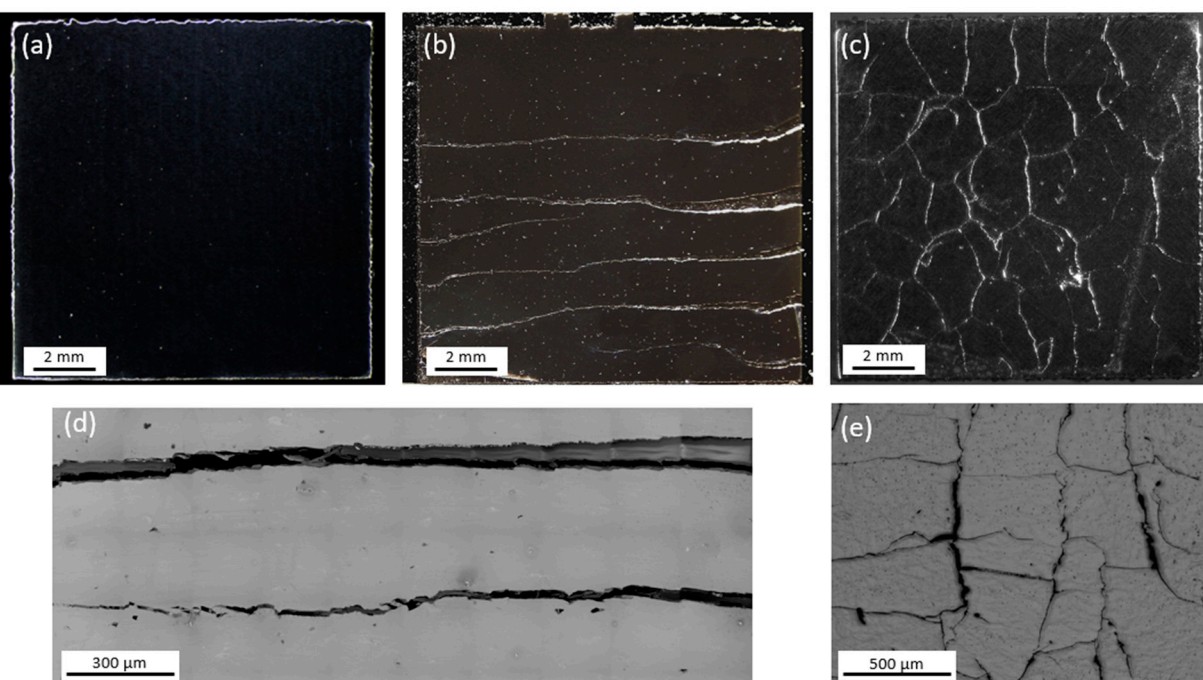

**Figure 7.** Stereomicrographs showing cross-sections of AlSi10Cu4Mg (**a**), AlSi10Cu8Mg (**b**), and AlCu20Si10Mg (**c**) samples. (**d**,**e**) Detailed SEM micrographs of AlSi10Cu8Mg delaminations and AlCu20Si10Mg hot cracks, respectively.

As can be observed in Figure 7a, the AlSi10Cu4Mg sample revealed a limited porosity and no cracks, achieving an adequate level of relative densification (99.4%, as reported in Table 2). On the contrary, in the AlSi10Cu8Mg and AlCu20Si10Mg systems (Figure 7b–e), unpredicted delaminations and cracks occurred when cubic samples were produced with the parameters suggested by the SST software. Looking closer at the AlSi10Cu8Mg system, we can see that the delaminations that occurred are macroscopic and perpendicular to the build direction (Figure 7d). Their morphology suggests that thermally induced stresses cause them during the layer-by-layer process. In PBF-LB/M, detrimental residual stresses could be thermally induced: compressive stresses induced in the cold solidified substrate that oppose the expansion of the warm layers involved in the laser scanning and tensile stresses formed during the cooling phase of the melted upper layers. Intense delaminations could be attributed to the thermal stress accumulation when exceeding the ultimate tensile strength of the AlSi10Cu8Mg alloy. These phenomena of residual stress accumulation and consequent delaminations and crack formation mainly affect alloys characterized by low thermal conductivity. Based on Aksöz et al.'s study on the Al-Cu binary system [26], it is reasonable to assume that the thermal conductivity decreases by increasing the Cu alloying content. This could explain the reason why these phenomena occurred in the AlSi10Cu8Mg system and not in the AlSi10Cu4Mg system. Several approaches can be taken to limit such phenomena, but a study previously conducted by the authors on the AlSi10Cu8Mg system demonstrated that only a synergetic approach involving a reduced scan speed (100 mm/s), high power (95 W), and the use of support structures is able to completely remove delamination, resulting in crack-free samples with a final relative density of 99.8% [28]. Moreover, the AlCu20Si10Mg system did not achieve a full densification due to the occurrence in the entire sample cross-section of severe cracks (Figure 7c). The morphology and distribution of these cracks differ significantly from the cracks observed in the AlSi10Cu8Mg sample. In fact, in the AlCu20Si10Mg sample, the cracks appear over the entire cross-section of the sample parallel and perpendicular to the building direction (Figure 7e). The cracks formed in the AlCu20Si10Mg sample are referred to as hot cracks, also known as hot tears or solidification cracks. Hot cracking typically occurs in the final stage of solidification, where the remaining liquid metal forms a thin film within the inter-dendritic regions of solidified

dendritic or columnar grains. At this point, the liquid film is unable to withstand the tensile strain resulting from the volumetric solidification shrinkage and thermal contraction [46]. The hot-cracking phenomenon can be traced to two causes: composition and, thus, specific alloy properties or processing conditions. Composition with high hot-cracking susceptibility often exhibits a broad solidification range, a relatively high thermal expansion coefficient, a high interdendritic liquid viscosity, and a poor liquid backfilling ability [47]. In addition, the PBF-LB/M process parameters play a key role in crack susceptibility. In particular, adequate energy density is required to avoid insufficient filling of the voids with liquid material during laser scanning, which could promote hot-crack formation [48]. To solve the high hot-cracking susceptibility observed in the AlCu20Si10Mg system from a process point of view, the same synergetic approach implemented on the AlSi10Cu8Mg system was applied to the AlCu20Si10Mg system, but this time exploring much lower scan speeds, up to 50 mm/s, and thus ensuring a 58% higher VED than the conditions suggested by the SSTs. However, this approach has led to a reduction in hot cracks, but their presence was still clearly visible and incompatible with a proper densification. The significantly low absorption rate recorded for the AlCu20SiMg system at the wavelength of the YAG laser could suggest the incompatibility of this composition to be processed and properly densified with the infrared laser, even optimizing process conditions. However, looking at the absorption rate pattern of AlCu20Si10Mg in Figure 5, it can be observed that, at shorter wavelengths, the laser absorbance reaches much higher values. Such considerations open the possibility of future process optimization studies with laser systems at lower wavelengths (green and blue lasers). However, before any advanced processability studies are undertaken, a preliminary investigation of the mechanical properties was necessary to understand the potential of high-copper Al-Si-Cu-Mg systems.

### 3.3. Influence of Copper on Microhardness

To understand how an incremental amount of alloyed Cu can affect the mechanical properties of the pre-alloyed AlSi10Cu4Mg, AlSi10Cu8Mg, and AlCu20Si10Mg systems, a preliminary investigation on microhardness values was conducted. By performing microhardness Vickers tests on entire cross-sectioned samples of AlSi10Cu4Mg and AlSi10Cu8Mg and in crack-free zones of AlCu20Si10Mg, and averaging ten measurements for each one, the mean microhardness values and standard deviations in Figure 8 were obtained.

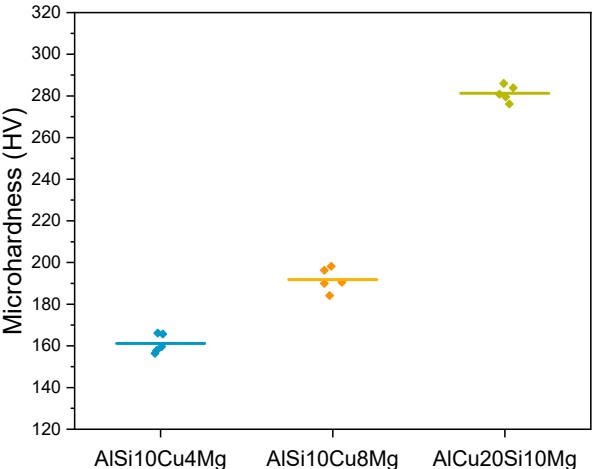

**Figure 8.** Microhardness results obtained by Vickers tests on AlSi10Cu4Mg, AlSi10Cu8Mg, and AlCu20Si10Mg as-built samples.

In line with the results obtained in a previous study by the authors, the AlSi10Cu8Mg system revealed a significant increase in microhardness compared to the benchmark AlSi10Cu4Mg system that was equal to 20%. Nonetheless, the high-copper AlCu20Si10Mg system demonstrated an impressive microhardness value, registering an increase of

67% over the benchmark. The mean microhardness value recorded by the AlCu20Si10Mg system (281 HV) reached a level never before achieved by PBF-LB/M Al-based systems in the as-built state. It is correct to point out that, although the AlCu20Si10Mg microhardness measurements were conducted in crack-free areas, hot cracks can still have a significant impact on the final mean value. Hot cracks are, in fact, present throughout the sample in the three directions (Figure 7c,e). It is therefore reasonable to assume that if some cracks were present underneath the indentation area, the material would appear softer, and a lower mean microhardness value would be recorded than those of properly densified material. Based on this assertion, it can be argued that the mean values recorded in this study for the AlCu20Si10Mg system underestimate the microhardness achievable after a densification optimization study of the PBF-LB/M process. Furthermore, considering that microhardness is related by a factor of 1/3 to the yield strength of a material, it is reasonable to expect that the AlCu20Si10Mg system subjected to tensile testing would show outstanding tensile strength [49]. However, a detailed microstructural analysis needs to justify this unprecedented result. Although it was not possible to obtain crack-free and fully densified AlCu20Si10Mg samples with the YAG laser, microstructural studies were carried out at the microscale level in the areas not involved in hot cracks.

### 3.4. Microstructural Investigations

The detailed SEM and FESEM analyses were evaluated to correlate the microhardness values recorded in pre-alloyed high-copper Al-Si-Cu-Mg systems with their microstructural features. These microstructural investigations were performed on samples treated with different chemical etchings to contrast specific microstructural features of the systems investigated. The Tucker etching was applied for a preliminary investigation to highlight the network features, as reported in Figure 9. In fact, the Tucker solution has already been used in the literature on Al-Si and Al-Cu systems to mainly etch the $\alpha$-Al phase, giving a strong contrast between the matrix and the phases enriching the network. As can be seen in Figure 9a, an SEM examination of AlSi10Cu4Mg-etched cross-sections parallel to the build direction revealed a network arrangement comprising two distinct regions. The first region, situated at the heart of the melt pool, emerged as a fine area, exhibiting smaller cells and more slender network walls. Adjacent to this area, along the melt pool boundaries (reported in yellow in Figure 9), an intricate network structure is visible that displays larger cells and enhanced thickness in the network walls. This distinction of melt pool areas remains clearly visible also in the AlSi10Cu8Mg sample of Figure 9b, but here the coarsen area along the melt pool boundary appears to be more extended. This effect could be attributed to the more energy-intensive process parameters and, thus, the significantly lower scan speed used to eliminate initial delaminations. Although the AlSi10Cu8Mg network morphology seems almost unaltered compared to the lower-copper system, an overall growth of the network in terms of cell enlargement and thickening of the arms could be detected. Conversely, the highest copper composition demonstrated a markedly different network morphology. In fact, the structure of the AlCu20Si10Mg network of Figure 9c appears coarsened, and the cells look shrunk, despite using more energy-intensive process parameters than those applied to the AlSi10Cu8Mg system.

In order to determine the origins of the different network morphologies, it is necessary to perform morphological analyses at higher magnifications and compositional analyses using BSE and EDS detectors. To achieve these advanced microstructural characterizations, it was decided to use the less intensive Keller etching that ensures a microstructure less hollow and more suited to BSE investigations and EDS analyses. Moving to higher magnifications with FESEM on AlSi10Cu4Mg samples (Figure 10), it can be observed that the network is formed of fine arms and more massive nodes characterized by a lamellar structure that is reminiscent of the eutectic Al-Si phase of the AlSi10Mg system.

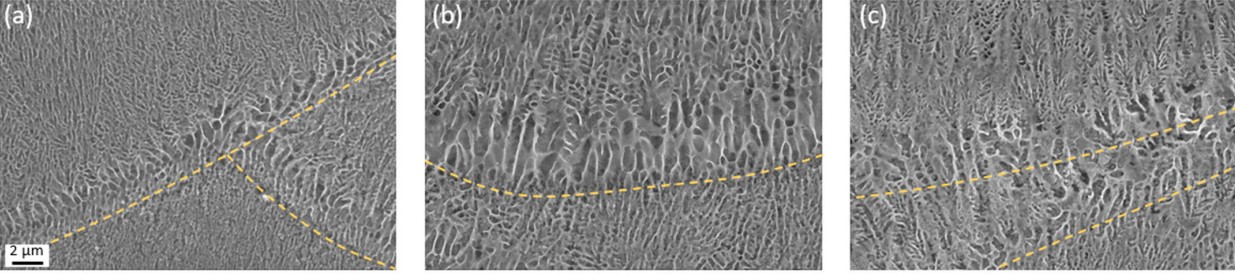

**Figure 9.** Micrographs of AlSi10Cu4Mg (**a**), AlSi10Cu8Mg (**b**), and AlCu20Si10Mg (**c**) samples etched with Tucker reagent.

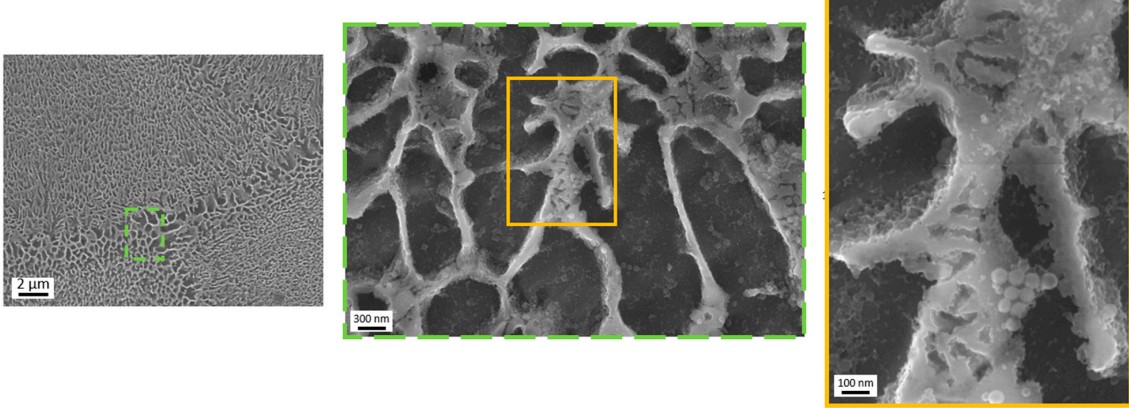

**Figure 10.** FESEM morphological investigation of the AlSi10Cu4Mg microstructure.

As already observed in the powders and bulk samples etched with Tucker solution, the network morphology of AlSi10Cu8Mg appeared very similar to the lower copper AlSi10Cu4Mg system, and for this reason, it is therefore not reported. On the contrary, the AlCu20Si10Mg system revealed a microstructure with important differences compared to the other systems. In particular, as summarized in Figure 11, the AlCu20Si10Mg microstructure appears very varied, showing regions with the cellular network, regions in which lamellar structures emerge in the coarse areas of the network (outlined in pink), regions with parallel needle-like dendritic structures (outlined in green), and regions with globular microstructures and flower-like dendritic structures (outlined in yellow).

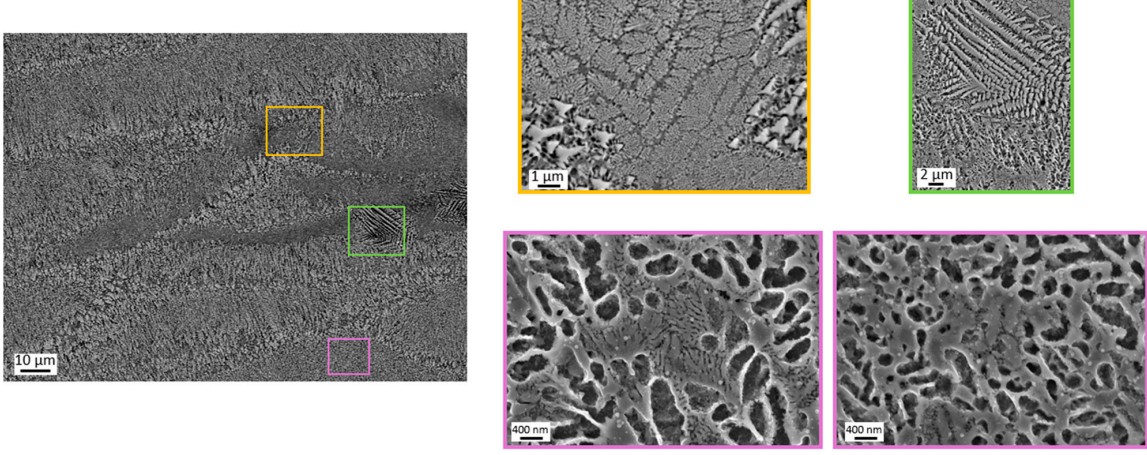

**Figure 11.** Varied microstructures of AlCu20Si10Mg observed with FESEM.

The huge presence of very fine and varied lamellar structures could be related to the impressive microhardness values measured in the AlCu20Si10Mg system. However, in order to better understand which dendritic morphologies have the greatest strengthening effect, future nanoindentation analyses have to be performed.

By exploiting the BSE detector, it was possible to observe the microstructure by distinguishing zones that are rich in heavy elements with a high atomic number and zones that are rich in light elements with a low atomic number (Figure 12). Relating to the Alsi10Cu4Mg and AlSI10Cu8Mg systems, the networks appeared to consist of two different phases: one with a higher atomic number (brighter in the BSE image) and one with a lower atomic number (darker in the BSE image). The SEM observation with the BSE detector also revealed that the heaviest elements are selectively arranged in the fine arms of the network and in the smaller nodes that do not present lamellar structures. Although the network morphology of AlSi10Cu8Mg appeared very similar to the lower copper AlSi10Cu4Mg systems, during the SEM observation with the BSE detector, a coarsening of the network and an increase in the copper-rich zones can be appreciated. The thickening of the strengthening network and the increased presence of copper-rich zones agrees with the increase in microhardness detected by the Vickers analysis with respect to the AlSi10Cu4Mg system. BSE observations for the AlCu20Si10Mg system were difficult due to the dense microstructural variety. EDS analyses were performed to better understand the observations performed with BSE detectors and to name the elements present in the networks of these systems.

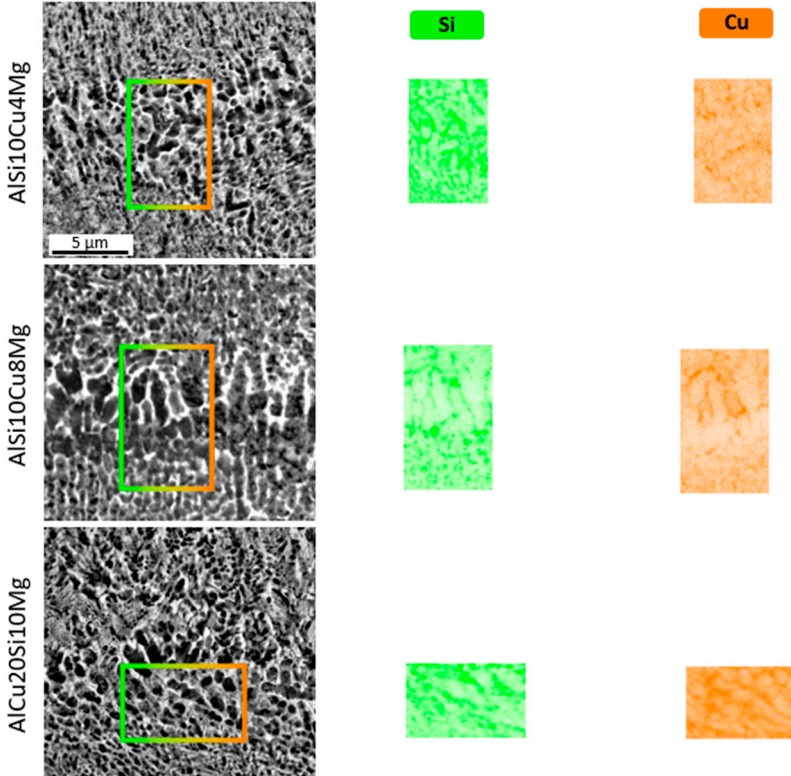

**Figure 12.** Cross-sectional micrographs of AlSi10Cu4Mg, AlSI10Cu8Mg, and AlCu20Si10Mg bulk samples observed by SEM with BSE detector and related EDS analyses.

EDS investigations determined the presence of different phases in the lattice, some rich in copper and others rich in silicon. However, establishing the exact stoichiometric ratio of the present phases was not possible through this analysis. A more specific analysis, XRD, was then used to identify the present phases and estimate the saturation condition of the Al matrices.

The spectra displayed in Figure 13 revealed the presence of fcc-Al, diamond cubic Si, and tetragonal Al$_2$Cu phase (θ phase). A study on the diffusion coefficients of several transition elements in the fcc Al revealed that Cu has one of the highest values [50] This leads to a higher probability of Al$_2$Cu precipitation during the rapid solidification. However, rapid solidification was proven to increase solubility limits and create supersaturated solutions. This consideration could explain the low intensity of the Al$_2$Cu phase peaks in the AlSi10Cu4Mg system and the gradual increase in intensity in the patterns of the higher copper content systems. Based on XRD patterns, Vegard's law was exploited to estimate the saturation level of these systems. This evaluation of AlSi10Cu4Mg, AlSi10Cu8Mg, and AlCu20Si10Mg resulted in a saturation value of Mg, Si, and Cu in the Al matrix of about 6.1, 7.7, and 8.0 at%, respectively. Assuming that Mg is entirely in solid solution and Si is in solid solution as in the AlSi10Mg system [51], the saturation value differences among the systems could be addressed by the different percentages of Cu in solid solution. By subtracting the assumed fixed contributions of Si and Mg from the global saturation levels obtained by applying Vegard's law and converting the values from at% to wt%, 3.9, 7.6, and 7.9 wt% were obtained as approximate values of copper in solid solution, respectively, for the AlSi10Cu4Mg, AlSi10Cu8Mg, and AlCu20Si10Mg systems. In contrast, the AlCu20Si10Mg system has less than half of the alloyed copper in the solid solution. Consequently, about 12 wt% of copper in the AlCu20Si10Mg alloy is outside the aluminum matrix and free to form other intermetallic phases. This assumption is also in line with the important morphological difference revealed by the microstructural analysis and the unknown peaks detected in the AlCu20Si10Mg XRD pattern. Furthermore, based on Vegard's law calculations, it can be stated that by processing Al-Si-Cu-Mg systems via PBF-LB/M, it is possible to obtain a supersaturated solution with a maximum copper concentration of slightly less than 8 wt%.

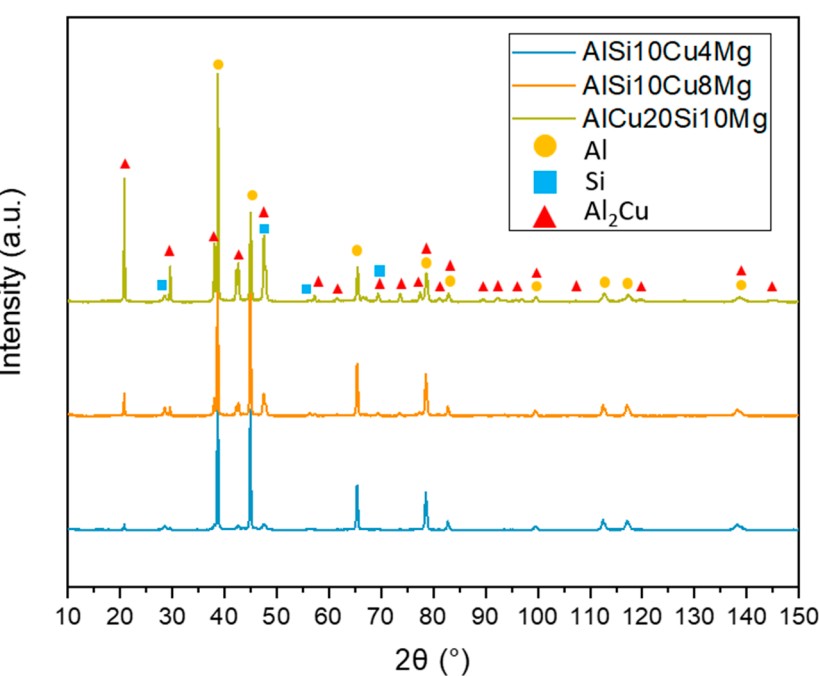

**Figure 13.** The XRD patterns of AlSi10Cu4Mg, AlSi10Cu8Mg, and AlCu20Si10Mg bulk samples and relative phase identification.

## 4. Conclusions

The present study aimed to explore high-copper Al-Si-Cu-Mg systems never before processed with PBF-LB/M. In particular, AlSi10Cu4Mg, AlSi10Cu8Mg, and AlCu20Si10Mg systems were investigated by analyzing the differences in PBF-LB/M processability, mi-

crostructural and compositional characteristics, and microhardness. The results obtained can be summarized as follows:

- The powders exhibited good flowability and packing, showing no different behavior with the increasing of the alloyed copper.
- The AlCu20Si10Mg system revealed a significantly lower laser absorbance level compared to the other two lower copper systems.
- The AlSi10Cu4Mg system was easily processed, obtaining fully dense and crack-free samples by using the process parameters suggested by the SSTs. On the other hand, systems with higher Cu content showed densification issues. The delaminations that occurred in the AlSi10Cu8Mg system were overcome by an accurate optimization of the process conditions. The hot cracks formed due to the poor laser absorbance of the AlCu20Si10Mg system were not eliminated by process optimization.
- Preliminary studies on the mechanical properties by microhardness measurements revealed unprecedented results, with an increase in mean microhardness values of 20 and 67% compared to the AlSi10Cu4Mg system for AlSi10Cu8Mg and AlCu20Si10Mg, respectively.
- Microstructural analyses revealed a marked morphological similarity between the AlSi10Cu4Mg and AlSi10Cu8Mg systems. In contrast, the AlCu20Si10Mg system revealed a very varied microstructure, with dendritic structures of different morphologies and a network that appears significantly enlarged with narrower cells.
- Through an accurate XRD pattern analysis and the application of Vegard's law, it was possible to estimate the Cu levels in solid solution. In particular, it has been proved that processing high-copper Al-Si-Cu-Mg alloys enables the formation of supersaturated systems with Cu levels slightly below 8 wt%.

Therefore, this study demonstrated how adding Cu beyond its solubility limit in Al leads to supersaturated solutions with important benefits for the mechanical properties of PBFed components. In particular, the peculiar microstructure and unprecedented microhardness values revealed in the AlCu20Si10Mg system might suggest a possible use of this composition in the as-built state without time- and energy-consuming heat treatment. To conclude, these encouraging results pave the way for future studies focusing on the optimization of PBF-LB/M processability and advanced mechanical and compositional characterizations of the AlCu20Si10Mg system.

**Author Contributions:** Conceptualization, M.L. and A.M.; methodology, A.M., M.L. and E.B.; investigation, A.M. and E.B.; writing—original draft preparation, A.M.; writing—review and editing, M.L. and E.B.; supervision, M.L. All authors have read and agreed to the published version of the manuscript.

**Funding:** This research received no external funding.

**Data Availability Statement:** Not applicable.

**Acknowledgments:** A special acknowledgement goes to Enrico Virgillito and Fabrizio Marinucci, who contributed to the powder gas atomization.

**Conflicts of Interest:** The authors declare no conflict of interest.

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
