# Peer review of "Effect of Cu Content on the PBF-LB/M Processing of the Promising Al-Si-Cu-Mg Composition"

_metals, doi:10.3390/met13071315_

Round 1

Reviewer 1 Report

1) "precipitation of dispersed Mg2Si nanoparticles [4]." was it really shown for the first time by the authors in 2017?

2) "solubilizing treatments" what does this term mean?

3) It is still not clear to the reader what the term PBF-LB / M means. Please enter it at the beginning of the article with a detailed description.

4) Perhaps I did not read carefully, but personally it is always not clear to me what concentration of chemical elements is hidden by the designations AlSi10Cu4Mg AlSi10Cu8Mg and AlCu20Si10Mg. For a simple understanding, in my opinion, it is necessary to designate the concentration of chemical elements with a table.

5) "Furthermore, considering that microhardness is related by a factor of 1/3 to the yield 455 strength of a material, it is reasonable to expect that the AlCu20Si10Mg system subjected 456 to tensile testing would show outstanding tensile strength." where did this pattern come from?

6) Conclusions need to be corrected. They should more concisely reflect the main scientific results, and not be a summary of the work done.

analysing,  summarised, behaviour, optimisation, characterised and other words and terms shoud be corrected

Reviewer 2 Report

The authors presented a study of LPBF fabricated Al-Si-Cu alloys with different Si/Cu/Mg contents. The following issues shodule be addressed. 

1. Fig. 4 can be replaced by Scheil solidification curves to explain the phase precipitation sequence as well as the following cracking susceptility in the later section.

2. The microcracks at higher magnifications should be further studied and added in Fig. 7.  

3. The effect of crckas on hardness value worth discussion.

4.  The correlation between microstructures and hardness should be established. 

5. The expression of line 552-553 is not appropriate as it can't explain the strong segregation evidenced by EDS analysis.

6. The conlcusion should be shortened, it is too long in the current form. 

Round 2

Reviewer 2 Report

The authors have answered my questions.